# Microbial Interactions in Nature: The Impact of Gram-Negative Bacilli on the Hyphal Growth of *Candida albicans*

**DOI:** 10.3390/pathogens14040327

**Published:** 2025-03-28

**Authors:** Madalina Adriana Bordea, Benjamin Thomas Georg Nutz, Alin-Dan Chiorean, Gabriel Samasca, Iulia Lupan, Laura Mihaela Simon, Lia Pepelea

**Affiliations:** 1Department of Microbiology, Iuliu Hatieganu University of Medicine and Pharmacy, 400151 Cluj-Napoca, Romania; bordea_madalina@yahoo.com (M.A.B.); benjaminnutz@gmail.com (B.T.G.N.); liapepelea@yahoo.com (L.P.); 2Departmernt of Molecular Biology, Iuliu Hatieganu University of Medicine and Pharmacy, 400151 Cluj-Napoca, Romania; chiorean.alin@yahoo.com; 3Department of Immunology, Iuliu Hatieganu University of Medicine and Pharmacy, 400151 Cluj-Napoca, Romania; 4Departmernt of Molecular Biology, Babes-Bolyai University, 400084 Cluj-Napoca, Romania; iulia.lupan@ubbcluj.ro

**Keywords:** *Candida albicans*, bacteria, interactions, therapeutic strategies

## Abstract

The escalating global prevalence of fungal and bacterial co-infections underscores the significant and multifaceted impact of ubiquitous microorganisms on both environmental equilibria and human well-being. The human microbiome, a complex ecosystem of bacterial communities, harbors opportunistic pathogens capable of inducing superinfections or concurrent infections with *Candida* spp. The intricate interplay, exemplified by the interaction between *Candida albicans* and diverse bacteria, necessitates rigorous investigation to elucidate mechanisms by which this polymicrobial behavior potentiates fungal virulence, particularly in immunocompromised individuals. Our study aims to comprehensively examine the ramifications of these interactions, with a specific focus on their influence on fungal virulence and the consequent exacerbation of disease severity. Achieving a comprehensive understanding of these complex relationships is paramount for informing effective clinical management strategies for infectious diseases, and the accurate identification of fungal–bacterial co-infections holds substantial implications for optimizing clinical treatment paradigms, especially in vulnerable immunocompromised hosts.

## 1. Introduction

Fungal infections pose a substantial clinical burden, affecting both immunocompromised patients and the wider population. Infections caused by *Candida albicans* are of particular concern due to their high prevalence and potential for severe morbidity. Candidiasis represents the most frequently observed fungal infection in humans. *C. albicans* is capable of causing a spectrum of infections, ranging from superficial cutaneous manifestations to life-threatening systemic conditions. This ability to colonize diverse host niches is underpinned by a broad repertoire of virulence factors and fitness attributes. Established virulence factors include the reversible morphological transition between yeast and hyphal states, surface expression of adhesins and invasins, thigmotropism, biofilm formation, and the secretion of hydrolytic enzymes. Furthermore, key fitness attributes enabling *C. albicans* survival and proliferation within the host environment include rapid adaptation to fluctuating environmental pH, metabolic flexibility, efficient nutrient acquisition systems, and robust stress response mechanisms [1].

The interplay between bacterial and fungal species, particularly Candida, is significantly influenced by secreted bacterial compounds that can either suppress or promote Candida’s growth and virulence. Understanding the mechanisms underlying these polymicrobial interactions is therefore crucial for the development of novel antifungal agents and for improving the efficacy of existing therapies. This intricate inter- and intra-kingdom crosstalk can modulate both the commensal and pathogenic states of bacteria and fungi. The bidirectional influence of Candida–bacteria interactions on virulence is highlighted by the observation that disruptions to microbial communities, such as those caused by antibiotic use or inflammation, can predispose individuals to Candida infections. Bacteria and fungi are found together in a myriad of environments and particularly in biofilms, where adherent species interact through diverse signaling mechanisms. Despite billions of years of coexistence, the area of research exploring fungal–bacterial interactions is still in its infancy. An example of a mutually beneficial interaction is coaggregation, a phenomenon that takes place in oral cavity biofilms where the adhesion of *C. albicans* to oral bacteria is very important for its colonization of the oral cavity. Not all *C. albicans*–bacteria interactions are mutually beneficial. In contrast, the interaction between *C. albicans* and *Pseudomonas aeruginosa* is described as being competitive and antagonistic in nature. *Pseudomonas aeruginosa*, a common Gram-negative soil bacterium and an opportunistic human pathogen, is well known for its ability to produce a blue phenazine, called pyocyanin, which is toxic to numerous bacteria and fungi. Recent reports indicate that *P. aeruginosa* and *Candida albicans* can coexist in a variety of different opportunistic infections, and a number of different molecular interactions between these two organisms have been described. Brand et al. showed that *P. aeruginosa* cells kill *C. albicans* hyphal cells by creating biofilms but not *C. albicans* yeast cells. They also investigated whether components of the hypha cell wall influenced susceptibility to the bacterium. It is well known that hypha-specific cell wall proteins (mannan components) of *C. albicans* are involved in adhesion and aggregation. His study demonstrated that mutant *C. albicans* strains that lacked the hypha-specific proteins Hyr1p, Hwp1p, and Als3p or enzymes involved in *N*-glycosylation (Och1p, Mnn4p, Mnt3p, Mnt4p, and Mnt5p) of surface glycoproteins were characterized by altered rates of killing by *P. aeruginosa* [2].

In several studies of direct interactions between *Pseudomonas* spp. and a *C. albicans* monomorphic *tup1* mutant that is constitutively hyphal, *P. aeruginosa* was found to form a biofilm on hyphae and to selectively kill them [3,4,5]. Suppression of fungal growth has also been correlated with production of the bacterial phenazine derivatives pyocynanin and 1-hydroxyphenazine in culture media. Recent studies have identified N-(3-oxododecanoyl)-l-homoserine lactone (HSL) as a primary quorum-sensing molecule in *P. aeruginosa*. This molecule plays a crucial role in regulating bacterial virulence factor production and has been shown to inhibit hyphal development, reduce biofilm formation, and induce apoptosis in *C. albicans* [6].

Cabral et al. found that *E. coli* kills *C. albicans* when co-cultured in vitro. This study suggests that this activity results from a soluble factor produced by *E. coli* in a manner that is independent of the presence of fungal cells. They also found that magnesium limitation is required for the observed toxicity. Another interaction is that occurring between *Staphylococcus aureus* and *C. albicans*, which, although not yet fully characterized, appears to be synergistic. These complex interactions would have significant clinical implications if they occurred in an immunocompromised host. Understanding the mechanisms of these interactions may lead to the development of novel antimicrobials [7,8,9].

In consequence, a thorough understanding of these complex interactions has the potential to inform the development of new therapeutic and preventative strategies targeting fungal infections [10]. Hence, our study investigated the impact of Gram-negative bacilli on the hyphal growth of *C. albicans*.

## 2. Materials and Methods

The experiment was conducted at the Department of Microbiology and the Cell Biology laboratory of Iuliu Hațieganu University of Medicine and Pharmacy Cluj-Napoca. Sterile, plastic inoculation loops were utilized to transfer samples of *C. albicans*, *Escherichia coli*, and *Pseudomonas aeruginosa* from stock cultures to agar plates. *C. albicans* is the most common pathogen in the majority of clinical settings in our geographic area.

Normal blood agar was employed as the growth medium to provide optimal nutrient conditions. Plates were subsequently incubated for 24 h at 37 °C under standard atmospheric conditions in a microbiology department incubator at Iuliu Hațieganu University of Medicine and Pharmacy. Following incubation, microbial growth was confirmed through microscopic examination of cellular morphology and macroscopic assessment of colony characteristics.

Specimens were collected using sterile cotton swabs from representative colonies exhibiting similar morphological characteristics for each cultivated organism. These organisms were then suspended in test tubes containing 0.9% NaCl saline solution (normal saline) and vortexed until a homogenous visual turbidity was achieved. A McFarland densitometer (bioSan DEN-1), calibrated to 0 McFarland using normal saline prior to each measurement, was utilized to determine the density of the suspensions. Saline solution was added incrementally to achieve a density of 0.5 McFarland for bacterial suspensions and 1 McFarland for *C. albicans* suspensions. This difference in density was implemented to account for the higher population density and smaller cell size characteristic of bacteria compared to the fungal cells of *C. albicans*. The aforementioned steps were repeated as needed to produce sufficient inoculum for all planned culture plates.

Subsequently, 100 μL of each inoculum suspension was applied to culture plates using precision pipettes according to the following experimental design:

I: Simple Agar Medium: 20 plates inoculated with *C. albicans* at 1 McFarland (control) 20 plates inoculated with *C. albicans* at 1 McFarland and *E. coli* at 0.5 McFarland 20 plates inoculated with *C. albicans* at 1 McFarland and *P. aeruginosa* at 0.5 McFarland;

II: Sabouraud Agar Medium: 20 plates inoculated with *C. albicans* at 1 McFarland (control) 20 plates inoculated with *C. albicans* at 1 McFarland and *E. coli* at 0.5 McFarland 20 plates inoculated with *C. albicans* at 1 McFarland and *P. aeruginosa* at 0.5 McFarland.

Following inoculation, plates were manually rotated, and sterile inoculation loops were employed to further disperse the organisms via a streaking technique. All plates were then incubated at 37 °C under normal atmospheric conditions for 120 h (5 days). After incubation, samples from three randomly selected, representative colonies of similar macromorphological appearance from each agar plate were transferred onto microscopy slides using sterile inoculation loops. Slides were examined under a light microscope without staining. A USB camera adapter was used to digitally capture microscopic images using ScopeImage 9.0 software for Microsoft Windows. Each slide was carefully examined for the presence of *C. albicans* hyphae using both simple and dark-field microscopy at 10×, 40×, and 100× magnification. Immersion oil was used for observations at 100× magnification. Qualitative morphological criteria were used to assign each agar plate a binary attribute of “Yes” or “No” based on the presence or absence of observed **C. albicans** pseudohyphae or hyphae. A plate was marked “Yes” if at least one pseudohyphal or hyphal formation was observed during microscopic examination and “No” if no such formations were observed after extensive searching at all magnifications. Finally, the resulting data were compiled into a listing table using Microsoft Excel Office16.

Our experiment has the limitations of an in vitro study. Microorganisms may behave differently in a lab setting than they would in their natural environment, which may limit the generalizability of the results. Another limitation is the number of samples. Additionally, this study does not explain the role of the virulence factors in these interactions. For this reason, further studies are needed.

## 3. Results

Statistical analysis was performed on the generated data, employing 2 × 2 contingency tables to represent observed results. To assess the relationship between Candida control group results and the two different bacterial species within each culture medium, Fisher’s exact test was applied to the corresponding 2 × 2 contingency tables. Fisher’s exact test is useful for contingency tables with very small sample sizes. Fisher’s exact test determines whether a statistically significant association exists between two categorical variables.

Visual representations, including graphs and charts presented in subsequent sections, were generated using GraphPad Prism version 9.5.1 for Microsoft Windows. All statistical tests were conducted using the same software. Microscopic examination of the 120 cultivated agar plates revealed the presence of pseudohyphal or hyphal formations on a total of 35 plates.

Of the 35 positive observations recorded, a substantial majority (31) originated from the 40 control cultures containing *Candida* spp. alone. Specifically, 14 positive observations were derived from samples cultured on simple agar control plates, while 17 were obtained from samples cultivated on Sabouraud agar control plates. A comparatively smaller number of positive observations (4) stemmed from cultures containing both fungal and bacterial populations. Further analysis of these mixed cultures revealed that 3 of the positive samples were isolated from the *C. albicans* and *E. coli* co-culture, with only a single positive observation noted in the multimicrobial culture containing fungal cells and *P. aeruginosa*.

This study investigated the impact of Gram-negative bacilli on the hyphal growth of *C. albicans*. The null hypothesis (H0) posited that the presence of Gram-negative bacilli in *C. albicans* cultures has no effect on hyphal growth. Conversely, the alternative hypothesis (H1) stated that the admixture of Gram-negative bacilli significantly affects hyphal growth (Figure 1 and Figure 2).

To assess these hypotheses, contingency tables were constructed to facilitate a direct statistical comparison between the number of positive results in the control group (pure *C. albicans*) and the experimental groups, which included *C. albicans* cultures co-inoculated with either *E. coli* or *P. aeruginosa*. Fisher’s exact test was employed to determine statistical significance, with a predetermined alpha level of 0.05. The calculated *p*-values for all four contingency tables were below the alpha level (*p* < 0.05), indicating statistical significance. Based on these findings, the null hypothesis (H0) was rejected. The data support the alternative hypothesis (H1), suggesting that the presence of Gram-negative bacilli has a significant effect on the hyphal growth of *C. albicans*. Representative images illustrating the observed differences between control and experimental groups on both culture media are presented in subsequent sections (Figure 3, Figure 4, Figure 5, Figure 6, Figure 7, Figure 8, Figure 9, Figure 10 and Figure 11).
(A)Control group images: exclusive *Candida albicans* culture.(B)Test group images: *Candida albicans* and either *E. coli* or *P. aeruginosa* polymicrobial culture.

Microscopic analysis revealed that the presence of bacteria significantly inhibited hyphal development in the fungal samples (*p* < 0.05). Furthermore, visual inspection of fungal cultures co-inhabiting with the two bacterial species on both agar types indicated a distinct shift in fungal cell distribution. In contrast to the singular distribution observed in the control samples, fungal cells in the presence of bacteria exhibited a clustered, tetracoccoid arrangement, as illustrated in the accompanying micrographs. This observation suggests a potential influence of bacterial presence on fungal cell morphology and organization.

## 4. Discussion

This experimental study investigated the potential influence of two common Gram-negative bacteria, *Escherichia coli* and *Pseudomonas aeruginosa*, on the in vitro hyphal development of *Candida albicans*. The objective was to determine whether the presence of these bacteria significantly alters the filamentous growth patterns of the fungal pathogen under controlled laboratory conditions. After statistical analysis led to the rejection of the null hypothesis, which posited no significant effect of Gram-negative bacteria on *Candida albicans* hyphal growth, subsequent observations indicated a negative correlation between co-culturing *C. albicans* with *Escherichia coli* and *Pseudomonas aeruginosa* and hyphal development in vitro. Of the 120 culture plates examined, hyphal growth was observed in only 35. A significant majority of these (31 out of 35) originated from the control cultures, suggesting that the presence of Gram-negative bacilli inhibits hyphal formation in *C. albicans* under experimental conditions.

Numerous explanations can be posited for this phenomenon, and a review of existing research literature reveals several plausible mechanisms. For instance, Morales et al. [11] identified candidacidal properties in *Pseudomonas* spp. Their research demonstrated that bacterial phenazines, secreted by strains of *P. aeruginosa*, inhibited fungal growth by targeting and eliminating germinated cells of *C. albicans*. Further supporting this antagonistic relationship, Salvatori et al. [12] reported that *Pseudomonas aeruginosa* bacteria also reduced filamentation of fungal cells. Brand et al. [2] showed that *P. aeruginosa* cells kill *C. albicans* hyphal cells by creating biofilms but not *C. albicans* yeast cells. They tested a range of mutants that lacked hypha-specific cell wall mannoproteins and others that lacked specific glycosyl epitopes. His study demonstrated that mutant *C. albicans* strains that lacked the hypha-specific proteins Hyr1p, Hwp1p, and Als3p or enzymes involved in *N*-glycosylation (Och1p, Mnn4p, Mnt3p, Mnt4p, and Mnt5p) of surface glycoproteins were characterized by altered rates of killing by *P. aeruginosa*.

However, the survival of *mnt1*Δ, *mnt2*Δ, and *mnt1*Δ/*mnt2*Δ mutants with truncated *O*-linked mannan was significantly reduced in the presence of *P. aeruginosa* as compared with the control strain, suggesting that *O*-mannan is protective against the *P. aeruginosa* killing activity. *P. aeruginosa* adhered preferentially to specific hyphae at all time points. There appeared to be no preferred attachment site relative to the length of the hypha, the presence or absence of a branch, or any other morphological parameter.

This observation highlights the antagonistic activity of these Gram-negative rods on Candida hyphal development, a finding consistent with the results observed in the present experiment.

Another factor influencing in vitro interaction is iron. A large proportion of the increased protein production, such as pyoverdine, was attributed to a siderophore, pyoverdine, specific to *P. aeruginosa.* This increase in pyoverdine is thought to be due to the increased iron utilization in the mixed biofilm. This was confirmed by the addition of iron, which abolished the production of pyoverdine. It was demonstrated that sequestration of available iron by pyoverdine results in decreased availability to *C. albicans*, although *C. albicans* is able to utilize iron bound to certain other microbial siderophores. Other evidence suggests that this phenomenon may not be of importance during in vivo interaction (in murine models). The authors suspect the heterogeneity of the biofilms between in vivo and in vitro studies may cause differential results. They also found that hypoxia influences the ability of *P. aeruginosa* to inhibit *C. albicans* filamentation in vitro compared to aerobic conditions. This was attributed to decreased AHL produced by *P. aeruginosa* in the presence of *C. albicans*. Additionally, the authors also speculated that competition for iron may also be greater during hypoxia. Therefore, both the interaction of *P. aeruginosa* with *C. albicans*, the concentration of oxygen, and iron competition influence the production of HSL [6].

Recent research has illuminated the complex interplay between bacteria and fungi, specifically focusing on the interaction between *Escherichia coli* and *Candida albicans*. A 2023 study by Bose et al. demonstrated that *E. coli* exhibits intrinsic antifungal activity against *C. albicans*. In vivo, the Gram-negative rod *E. coli* inhibited the growth of Candida cells in animal models co-injected with both microorganisms. This antifungal effect is hypothesized to be mediated by an unidentified metabolite produced by *E. coli*. The observed inhibition of *C. albicans* growth in the current experiment may also be attributable to this same bacterial-derived molecule. Conversely, Bose et al. reported that the Gram-positive coccus *Staphylococcus aureus* appears to promote hyphal development in *C. albicans*, suggesting a mutualistic relationship between the two organisms. Given these contrasting effects, further investigation into the interaction between *S. aureus* and *C. albicans* is warranted to elucidate the mechanisms underlying the differential influences of Gram-positive and Gram-negative bacteria on fungal morphology and development [13].

*Candida albicans* and *Staphylococcus* spp. represent the predominant fungal and bacterial pathogens, respectively, implicated in bloodstream infections globally. Notably, approximately 20% of *C. albicans* bloodstream infections present as polymicrobial, with *Staphylococcus epidermidis* and *Staphylococcus aureus* identified as the most and third most frequently co-isolated organisms, respectively. These microorganisms are recognized for their capacity to establish persistent biofilms within the host environment. Interactions occurring within these biofilm communities can potentiate virulence, drug tolerance, and immune evasion. These factors collectively contribute to an adverse impact on morbidity and infection prognosis, frequently resulting in elevated mortality rates. Consequently, a comprehensive characterization of the interactions between these species is warranted to elucidate their roles in influencing morbidity and mortality outcomes [14,15,16,17].

In the mixed fungal–bacterial cultures examined in this study, *Candida albicans* cells were observed to aggregate, frequently forming clusters of two or four cells. This contrasted with the Candida-only control cultures, where cells were predominantly found as singular entities, except during hyphal or pseudohyphal development. We hypothesize that this clustering behavior is attributable to the presence of bacteria within the shared environment. While the underlying mechanisms remain unclear, further experimental investigation and a comprehensive review of relevant literature are warranted to elucidate this phenomenon. Potential explanations for the observed aggregation include, but are not limited to, a defensive response by the fungal cells to bacterial secretions, interspecific competition for resources, or the initiation of biofilm formation. Alternative explanations, such as insufficient sample dilution or excessively high initial fungal cell concentrations (approximating or exceeding 1 McFarland standard) prior to agar plate inoculation, should also be considered. While this study provides initial insights, it is crucial to acknowledge its limitations and emphasize that the findings cannot be considered fully representative of in vivo conditions. Several factors necessitate further investigation and methodological refinement in future research iterations. Primarily, the relatively small sample size, constrained by resource limitations, the study’s scope, time constraints, and the limited personnel involved, compromises the statistical power and generalizability of the results. A more comprehensive study, incorporating a significantly larger number of samples (e.g., experimental plates), is essential to enhance the robustness and statistical significance of future findings. Further research should expand the scope of this study by incorporating a broader spectrum of bacterial species. Future investigations could also explore the differential effects of Gram-negative bacteria on hyphal development across a range of fungal pathogens relevant to human health. Furthermore, the inclusion of diverse Gram-positive bacterial species would provide a more comprehensive understanding of bacteria–fungi interactions. One potential limitation of the experimental design lies in the possibility of undetected growth formations on certain culture plates. The random sampling methodology employed for microscopic observation introduces the risk that localized hyphal growth, present on some plates, may have been missed during sample extraction. This could result in false negative findings, where hyphal growth was actually present but not observed. Conversely, the likelihood of false positives is considered minimal, as samples exhibiting positive observations are highly likely to accurately represent the presence of hyphal growth on the corresponding culture plate. In the current study, standard mycological culture techniques utilizing Sabouraud’s dextrose agar proved adequate for in vitro cultivation. However, future research could explore alternative culture media and environmental conditions to potentially enhance hyphal development and provide a more comprehensive understanding of fungal growth. Specifically, inoculation on specialized media, such as Lees medium, which is known for its enhanced support of fungal proliferation, should be considered. Furthermore, investigating the influence of microaerophilic environments with elevated CO_2_ levels could elucidate whether varying atmospheric conditions significantly impact hyphal growth patterns and subsequent experimental outcomes. These refinements to culture methodology may contribute to more robust and informative results in future studies. In this study, microscopic sample preparation and evaluation were conducted without the use of staining techniques. Unstained samples provided sufficient visual information to ascertain the presence of target microorganisms and to identify the presence or absence of hyphal or pseudohyphal structures, thereby addressing the research question for each sample. However, to enhance future microscopic analyses and mitigate the potential for false negative results, the application of staining methods such as Gram staining, Periodic Acid–Schiff (PAS) staining, or Grocott’s methenamine silver (GMS) stain for fungi should be considered. These techniques may provide improved visualization and facilitate more accurate identification of microbial features. Acknowledging the aforementioned limitations, future research is imperative to ascertain the precise etiology of the observed outcomes. This should encompass rigorous error correction and a refinement of the experimental design. Such enhancements are crucial to replicate the findings and foster a deeper understanding of the intricate multimicrobial interactions occurring between *Candida albicans* and Gram-negative bacilli.

## 5. Conclusions

Based on the results of our in vitro experiment, we rejected the null hypothesis, which posited that the admixture of Gram-negative bacilli to *C. albicans* cultures has no effect on hyphal growth. Consequently, we conclude that the alternative hypothesis, stating that the admixture of Gram-negative bacilli significantly affects hyphal growth, is a more accurate representation of the observed phenomenon within our limited sample. Specifically, the effect appears to be inhibitory on hyphal development. Microscopic examination of the cultured samples revealed a reduced frequency of pseudohyphal and hyphal growth in multi-microbial cultures containing both *C. albicans* and *E. coli* or *P. aeruginosa* in approximately equal proportions. This was in comparison to control cultures where *C. albicans* was cultured alone for 120 h under standard atmospheric conditions at 37 °C on both simple agar and Sabouraud agar. Furthermore, the cohabitation with Gram-negative bacilli seemed to promote the formation of clustered *Candida albicans* cells, typically arranged in groups of four. Because of the heterogeneity of the biofilms between in vivo (murine models) and in vitro experiments, further studies are needed.

## Figures and Tables

**Figure 1 pathogens-14-00327-f001:**
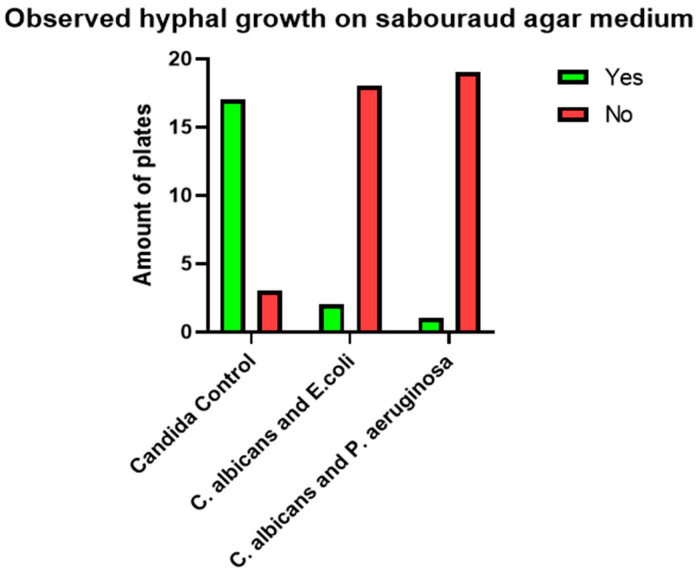
Hyphal growth in the presence of Gram-negative bacilli.

**Figure 2 pathogens-14-00327-f002:**
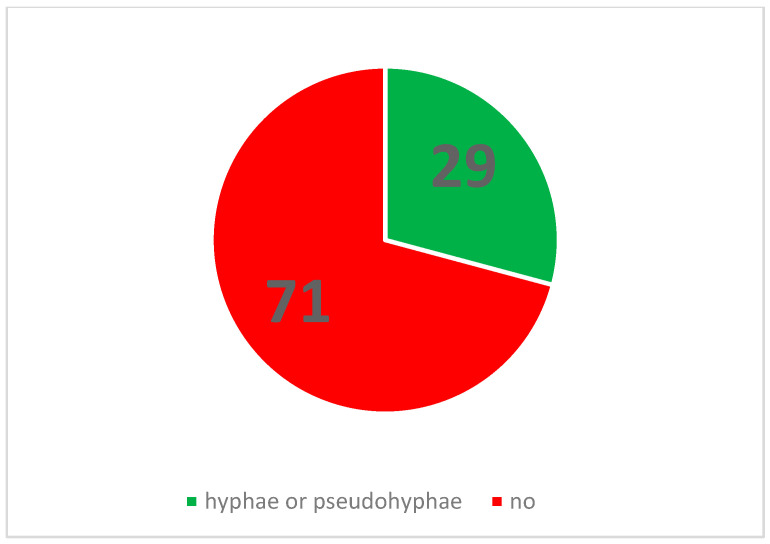
Proportion of numbers of observed plates featuring hyphal or pseudohyphal growth to negative plates.

**Figure 3 pathogens-14-00327-f003:**
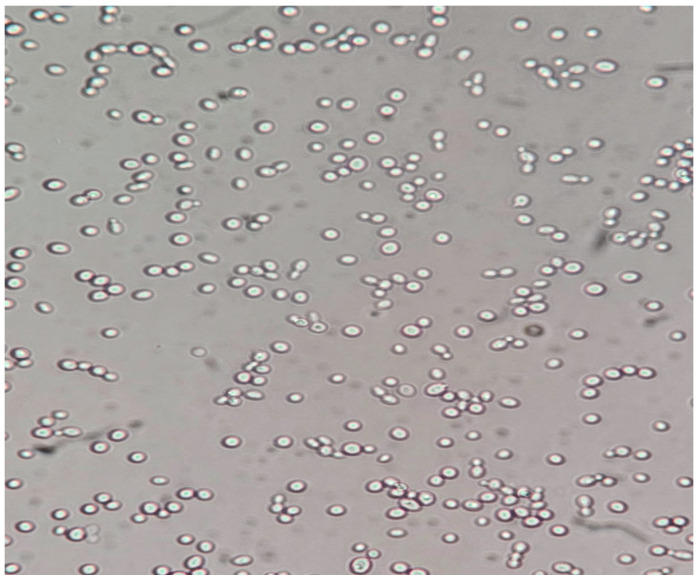
*Candida albicans* yeast cells from a sample on simple agar plate #2 after 120 h of incubation at 37 °C under normal atmospheric conditions. 40× magnification, no staining.

**Figure 4 pathogens-14-00327-f004:**
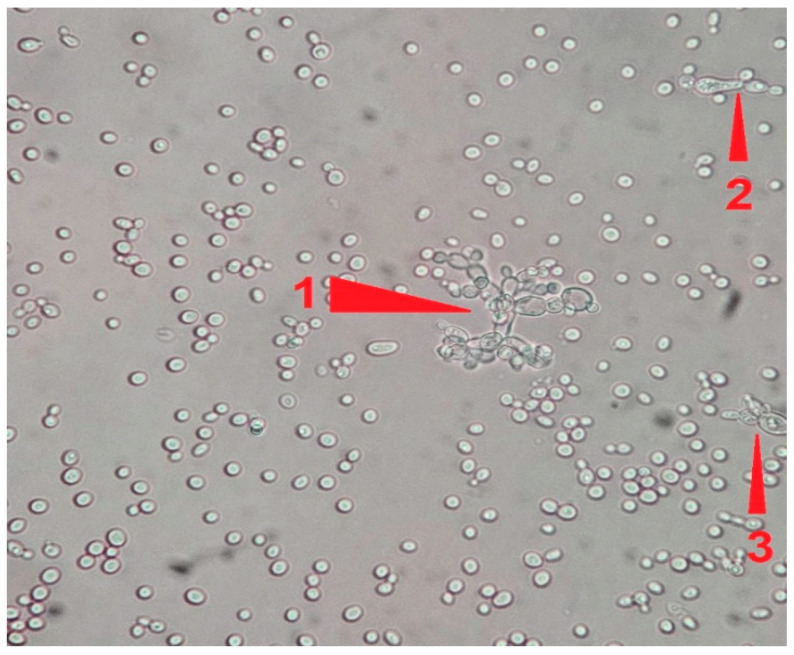
1: Large *Candida albicans* pseudohyphal formation with multiple ramifications. 2: Budding yeast cells in the process of pseudohyphal formation. 3: Smaller pseudohyphal formation in the process of enlargement. From a sample on Sabouraud agar plate #15 after 120 h of incubation at 37 °C under normal atmospheric conditions. 40× magnification, no staining.

**Figure 5 pathogens-14-00327-f005:**
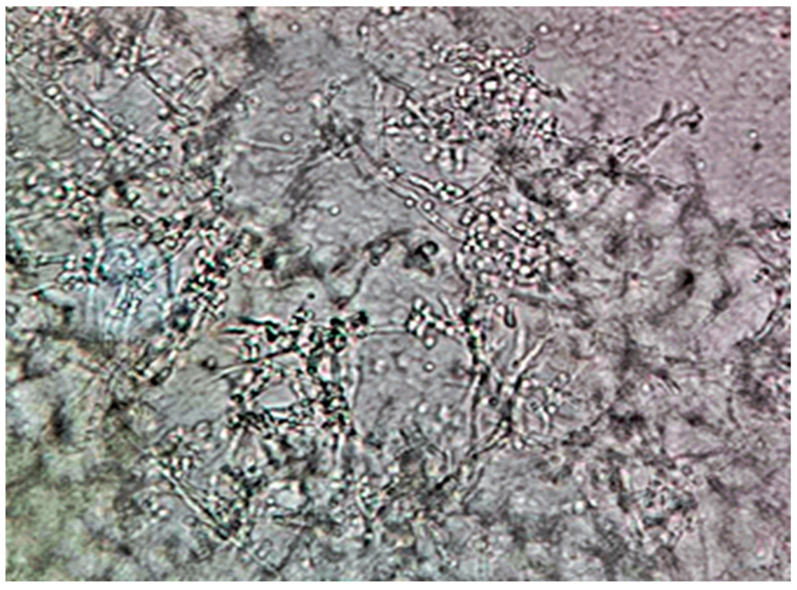
*Candida albicans* hyphal network from sample on Sabouraud agar plate #15 after 120 h of incubation at 37 °C under normal atmospheric conditions. 40× magnification, no staining.

**Figure 6 pathogens-14-00327-f006:**
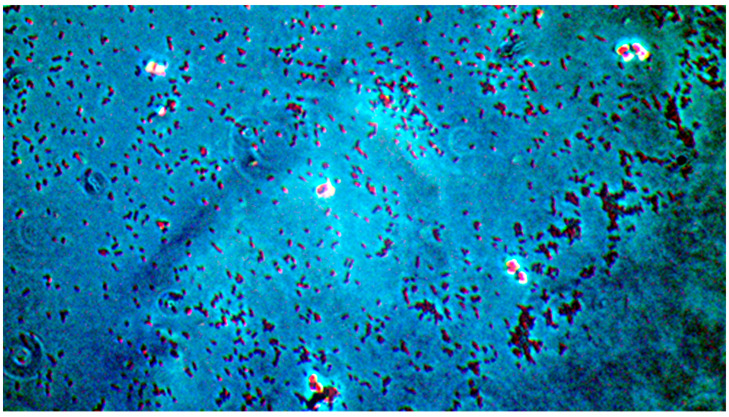
*Candida albicans* yeast cell clusters and *P. aeruginosa* cells at 40× magnification seen on simple agar medium plate #5 after 120 h of incubation at 37 °C under normal atmospheric conditions, no staining.

**Figure 7 pathogens-14-00327-f007:**
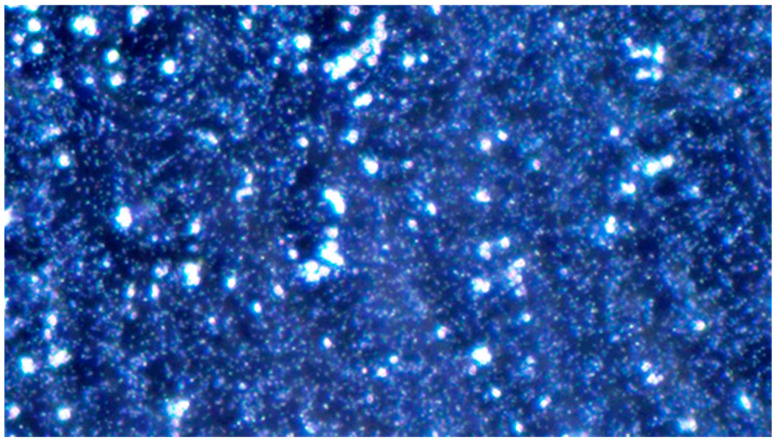
*Candida albicans* yeast cell clusters between *P. aeruginosa* cells at 20× magnification darkfield microscopy seen on simple agar medium plate #19 after 120 h of incubation at 37 °C normal atmospheric conditions with no staining.

**Figure 8 pathogens-14-00327-f008:**
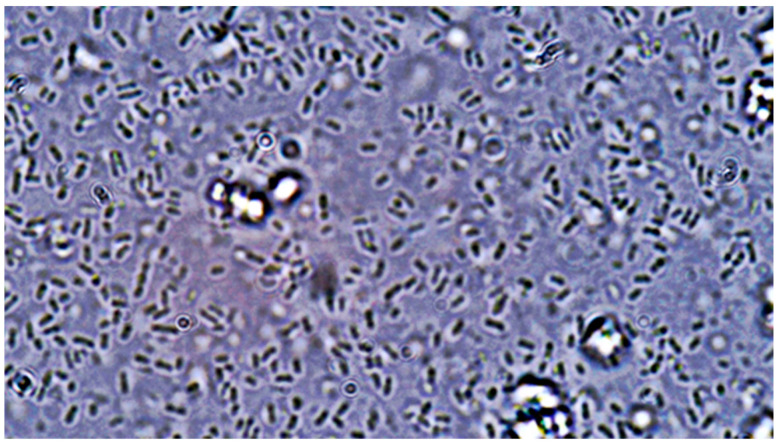
*Candida albicans* yeast cell clusters between *P. aeruginosa* cells at 100× magnification darkfield microscopy seen on simple agar medium plate #19 after 120 h of incubation at 37 °C under normal atmospheric conditions with no staining.

**Figure 9 pathogens-14-00327-f009:**
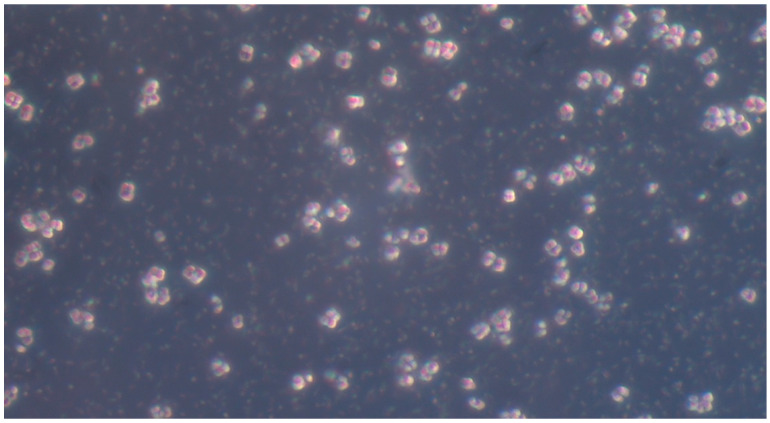
*Candida albicans* yeast cell clusters between *E. coli* cells at 40× magnification seen on simple agar medium plate #20 after 120 h of incubation at 37 °C under normal atmospheric conditions with no staining.

**Figure 10 pathogens-14-00327-f010:**
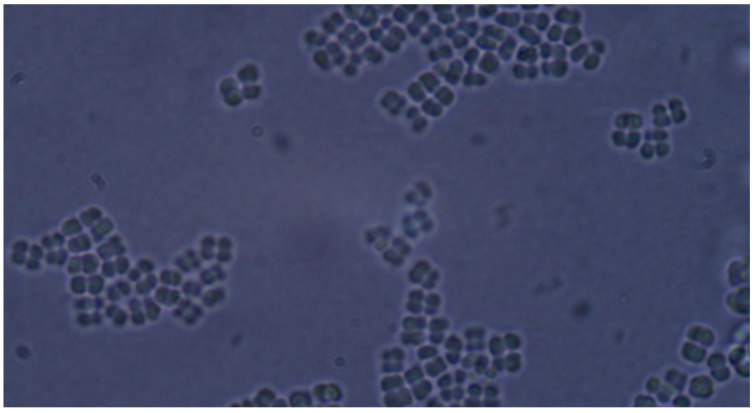
*Candida albicans* yeast cell clusters between *E. coli* cells at 100× magnification seen on simple agar medium plate #10 after 120 h of incubation at 37 °C under normal atmospheric conditions with no staining.

**Figure 11 pathogens-14-00327-f011:**
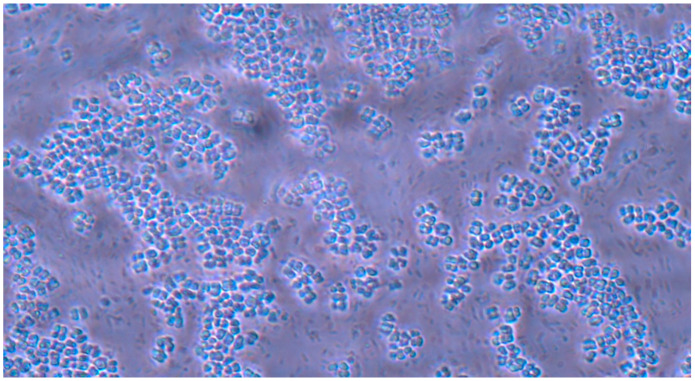
*Candida albicans* yeast cell clusters between *E. coli* cells at 40× magnification seen on simple agar medium plate #12 after 120 h of incubation at 37 °C under normal atmospheric conditions with no staining.

## Data Availability

The original contributions presented in this study are included in the article. Further inquiries can be directed to the corresponding author.

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
