# Peer review of "Microbial Interactions in Nature: The Impact of Gram-Negative Bacilli on the Hyphal Growth of Candida albicans"

_pathogens, 2025, doi:10.3390/pathogens14040327_

Round 1
Reviewer 1 Report
Comments and Suggestions for Authors
Introduction
The introduction is very concise and is based only on review articles. It should be expanded and bring more examples from scientific literature on the interaction between fungi and bacteria.
In the current version, the bacteria are presented in general terms only. Therefore, I suggest to specify and providing examples which types of bacteria are involved in the mutual infection. Specifically provide examples from the literature, regarding the interaction between Candida albicans and Escherichia coli & Pseudomonas aeruginosa that you used in this article.
Materials and Methods
The methodology is based on very basic methods that do not give any additional information on how the bacteria affect the fungi. It only shows that the phenomenon exists and this finding is already known in the literature.
Specific comments
Line 63 – write C. albicans
Line 78 – write C. albicans
Line 127 – write C. albicans
Line 135 – write statistical
Line 197 – write significantly
Line 205– write C. albicans
Line 279 - CO2 write CO2
Only give a full name of the species the first time you mention it and write the names in italic.
Author Response
Reviewer 1 :
Introduction
The introduction is very concise and is based only on review articles. It should be expanded and bring more examples from scientific literature on the interaction between fungi and bacteria.
In the current version, the bacteria are presented in general terms only. Therefore, I suggest to specify and providing examples which types of bacteria are involved in the mutual infection. Specifically provide examples from the literature, regarding the interaction between Candida albicans and Escherichia coli & Pseudomonas aeruginosa that you used in this article.
Answer :
Bacteria and fungi are found together in a myriad of environmentas and particularly in biofilms, where adherent species interact through diverse signaling mechanisms. Despite billions of years of coexistence, the area of research exploring fungal-bacterial interactions is still in its infancy. An example of a mutually beneficial interaction is coaggregation, a phenomenon that takes place in oral cavity biofilms where the adhesion of C.albicans to oral bacteria is very important for its colonization of the oral cavity. Not all C. albicans-bacteria interactions are mutually beneficial. In contrast, the interaction between C.albicans and Pseudomonas aeruginosa is described as being competitive and antagonistic in nature. Pseudomonas aeruginosa, a common Gram-negative soil bacterium and an opportunistic human pathogen, is well known for its ability to produce a blue phenazine, called pyocyanin, which is toxic to numerous bacteria and fungi. Recent reports indicate that P. Aeruginosa and Candida albicans can coexist in a variety of different opportunistic infections and a number of different molecular interactions between these two organisms have been described. Cabral et al.found that E. coli kills C. albicans when co-cultured in vitro. The study suggests that this activity results from a soluble factor produced by E. coli in a manner that is independent of the presence of fungal cells. They also found that magnesium limitation is required for the observed toxicity. Another interaction is that occurring between Staphyloococcus aureus and C.albicans, which although not yet fully characterized, appears to be synergistic. These complex interactions would have significant clinical implications if they occured in an immunocompromised host.Understanding the mechanisms of these interactions may lead to the development of novel antimicrobials.
Specific comments
Line 63 – write C. albicans
Line 78 – write C. albicans
Line 127 – write C. albicans
Line 135 – write statistical
Line 197 – write significantly
Line 205– write C. albicans
Line 279 - CO2 write CO2
Answer: Lines are solved.
I appreciate your comments. Thank you for your time!
Reviewer 2 Report
Comments and Suggestions for Authors
The author should address the following comments:
-
The title needs revision.
-
The abstract and introduction lack sufficient information about the objectives.
-
Both the abstract and introduction appear to be AI-generated.
-
The methodology is not detailed or reproducible. Since Candida albicans typically exhibits an antagonistic relationship with Pseudomonas aeruginosa, this interaction should be discussed by the author.
-
Several studies have already been published on the proposed topic.
-
Initially, the author should present an image showing Candida albicans hypha formation. In Figure 3, there is only a single Candida albicans hypha, but the image should demonstrate 100% hyphal involvement, after which the author can assess the influence of bacteria on hyphal production.
-
The author should quantify the presence of Candida albicans and bacterial cells in the mixed culture using the colony counting method and report the percentage of co-occurrence of both cell types in the culture.
- There is no statistical analysis of data in Figure 1.
Author Response
Reviewer 2:
1. The title needs revision.
Answer : The title has been changed
Microbial Interactions in Nature: the impact of Gram-negative bacilli on the hyphal growth of C.albicans.
2. The abstract and introduction lack sufficient information about the objectives.
3. Both the abstract and introduction appear to be AI-generated.
4. The methodology is not detailed or reproducible. Since Candida albicans typically exhibits an antagonistic relationship with Pseudomonas aeruginosa, this interaction should be discussed by the author.
5. Several studies have already been published on the proposed topic.
Answer :
More data are added
: Bacteria and fungi are found together in a myriad of environmentas and particularly in biofilms, where adherent species interact through diverse signaling mechanisms. Despite billions of years of coexistence, the area of research exploring fungal-bacterial interactions is still in its infancy. An example of a mutually beneficial interaction is coaggregation, a phenomenon that takes place in oral cavity biofilms where the adhesion of C.albicans to oral bacteria is very important for its colonization of the oral cavity. Not all C. albicans-bacteria interactions are mutually beneficial. In contrast, the interaction between C.albicans and Pseudomonas aeruginosa is described as being competitive and antagonistic in nature. Pseudomonas aeruginosa, a common Gram-negative soil bacterium and an opportunistic human pathogen, is well known for its ability to produce a blue phenazine, called pyocyanin, which is toxic to numerous bacteria and fungi. Recent reports indicate that P. Aeruginosa and Candida albicans can coexist in a variety of different opportunistic infections and a number of different molecular interactions between these two organisms have been described. Cabral et al.found that E. coli kills C. albicans when co-cultured in vitro. The study suggests that this activity results from a soluble factor produced by E. coli in a manner that is independent of the presence of fungal cells. They also found that magnesium limitation is required for the observed toxicity. Another interaction is that occurring between Staphyloococcus aureus and C.albicans, which although not yet fully characterized, appears to be synergistic. These complex interactions would have significant clinical implications if they occured in an immunocompromised host.Understanding the mechanisms of these interactions may lead to the development of novel antimicrobials.
6. Initially, the author should present an image showing Candida albicans hypha formation. In Figure 3, there is only a single Candida albicans hypha, but the image should demonstrate 100% hyphal involvement, after which the author can assess the influence of bacteria on hyphal production.
Answer :
Figure 3 and 4 have been changed.In the new format these figures are 4 and 5.
7. There is no statistical analysis of data in Figure 1.
Answer :
Figure 1 has been improved with a new figure (figure 2).
I appreciate your comments. Thank you for your time!
Reviewer 3 Report
Comments and Suggestions for Authors
In an in vitro study, Bordea et al. investigated the impact of Gram-negative co-infection on Candida hyphal production.
Both E. coli and, primarily, P. aeruginosa significantly increased hyphal production.
As I am not a mycologist, I will limit my discussion to the presentation, the underlying hypothesis, and the conclusions, without challenging the experimental procedure.
Concern: The pathophysiology behind the increased hyphal production is not explained.
More generally, the context is not clearly provided, and the existing knowledge regarding Candida/bacterial interactions is not sufficiently detailed. To enrich the discussion, I suggest including relevant articles on virulence factors associated with Candida-bacterial interactions. Below are some suggestions:
- Many important studies are not cited.
- The hypotheses are not clearly stated.
- A more extensive review could improve the manuscript. I have listed some references (non-exhaustive) for consideration:
PMID: 2416659
PMID: 23622953
PMID: 21689424
PMID: 28114348
PMID: 16424420
PMID: 31020361
Additionally, the following articles are crucial for peritonitis research:
Peters et al., Infect Immun 2013; and Nash et al., Infect Immun 2014.
The limitations of the experiments should be discussed in the context of the available data.
Minor note: The purpose of the study, mentioned in line 126, should ideally be placed at the end of the introduction.
Comments on the Quality of English Languagenone
Author Response
Reviewer 3:
Comments and Suggestions for Authors
In an in vitro study, Bordea et al. investigated the impact of Gram-negative co-infection on Candida hyphal production.
Both E. coli and, primarily, P. aeruginosa significantly increased hyphal production.
As I am not a mycologist, I will limit my discussion to the presentation, the underlying hypothesis, and the conclusions, without challenging the experimental procedure.
Concern: The pathophysiology behind the increased hyphal production is not explained.
More generally, the context is not clearly provided, and the existing knowledge regarding Candida/bacterial interactions is not sufficiently detailed. To enrich the discussion, I suggest including relevant articles on virulence factors associated with Candida-bacterial interactions. Below are some suggestions:
- Many important studies are not cited.
- The hypotheses are not clearly stated.
- A more extensive review could improve the manuscriptAdditionally, the following articles are crucial for peritonitis research:
Peters et al., Infect Immun 2013; and Nash et al., Infect Immun 2014.
Answer :
More data are added:
Bacteria and fungi are found together in a myriad of environmentas and particularly in biofilms, where adherent species interact through diverse signaling mechanisms. Despite billions of years of coexistence, the area of research exploring fungal-bacterial interactions is still in its infancy. An example of a mutually beneficial interaction is coaggregation, a phenomenon that takes place in oral cavity biofilms where the adhesion of C.albicans to oral bacteria is very important for its colonization of the oral cavity. Not all C. albicans-bacteria interactions are mutually beneficial. In contrast, the interaction between C.albicans and Pseudomonas aeruginosa is described as being competitive and antagonistic in nature. Pseudomonas aeruginosa, a common Gram-negative soil bacterium and an opportunistic human pathogen, is well known for its ability to produce a blue phenazine, called pyocyanin, which is toxic to numerous bacteria and fungi. Recent reports indicate that P. Aeruginosa and Candida albicans can coexist in a variety of different opportunistic infections and a number of different molecular interactions between these two organisms have been described. Cabral et al.found that E. coli kills C. albicans when co-cultured in vitro. The study suggests that this activity results from a soluble factor produced by E. coli in a manner that is independent of the presence of fungal cells. They also found that magnesium limitation is required for the observed toxicity. Another interaction is that occurring between Staphyloococcus aureus and C.albicans, which although not yet fully characterized, appears to be synergistic. These complex interactions would have significant clinical implications if they occured in an immunocompromised host.Understanding the mechanisms of these interactions may lead to the development of novel antimicrobials.
The limitations of the experiments should be discussed in the context of the available data.
Answer :Limitations are inserted at Material and Method
Our experiment has the limitations of an in vitro study. Microorganisms may behave differently in a lab setting than they would in their natural environment, which may limit the generalizability of the results. Another limitation is the number of samples. Additionally, the study does not explain the role of the virulance factors in these interactions. For this reason further studies are needed.
Minor note: The purpose of the study, mentioned in line 126, should ideally be placed at the end of the introduction.
Answer: The purpose of the study has been placed at the end of the introduction.
Our study investigated the impact of Gram-negative bacilli on the hyphal growth of C. albicans.
I appreciate your comments. Thank you for your time!
Reviewer 4 Report
Comments and Suggestions for Authors
I believe that if the objective is reformulated and the changes are made, it may represent a good contribution to the scientific community.
- The title of article does not represent the importance of the study, I think the authors should improve it according to the reformulation of the objective.
2. The objective should be reformulated, since it is not only to determine whether or not it inhibits the formation of hyphae and pseudohyphae, but also to include the importance of their inhibition by Gram-negative bacilli.
It is important that the authors present the statistical results in table or tables, so that they are more understandable.
4. Explain why Fisher's test was used or chosen.
5. The methodology section should be clearer in the development of all the processes that were carried out.
6. Something interesting that the authors should take into account and explain in the results is why they used C. albicans and not other species of this genus. The latter is related to the fact that C. albicans is one of the two species that produce pseudohyphae and hyphae together with C. tropicalis. They should also explain in the methodology how they determined that the yeasts were C. albicans and not other species of the genus, since they mention Candida spp. in a section. Here they can preferably place the means used for identification (preferably Proteomics or Molecular Biology).
7. The authors should explain figure No. 5 in a clearer way.
8. In Figure 6, arrows should be used to indicate where the C. albicans yeasts are and where the P. aeruginosa bacteria are. - 9. Both figures 7 and 9 are of very poor quality, it has a bad resolution or bad focus, so I suggest to improve or change it.
10. It is not clear to me the objective of the study, at the beginning of the article.
11. I understand that in the discussion they talk about the importance of the inhibition of C. albicans by compounds or metabolites released by Gram-negative bacilli (E. coli and P. aeruginosa), but this is not specifically mentioned in the introduction and conclusions, so I suggest improving them.
I think the English is of good quality
Author Response
· Reviewer 4:
Comments and Suggestions for Authors
I believe that if the objective is reformulated and the changes are made, it may represent a good contribution to the scientific community.
1. The title of article does not represent the importance of the study, I think the authors should improve it according to the reformulation of the objective.
· Answer : The title has been changed
· Microbial Interactions in Nature: the impact of Gram-negative bacilli on the hyphal growth of C.albicans.
2. The objective should be reformulated, since it is not only to determine whether or not it inhibits the formation of hyphae and pseudohyphae, but also to include the importance of their inhibition by Gram-negative bacilli.
Answer :
More data are added:
Bacteria and fungi are found together in a myriad of environmentas and particularly in biofilms, where adherent species interact through diverse signaling mechanisms. Despite billions of years of coexistence, the area of research exploring fungal-bacterial interactions is still in its infancy. An example of a mutually beneficial interaction is coaggregation, a phenomenon that takes place in oral cavity biofilms where the adhesion of C.albicans to oral bacteria is very important for its colonization of the oral cavity. Not all C. albicans-bacteria interactions are mutually beneficial. In contrast, the interaction between C.albicans and Pseudomonas aeruginosa is described as being competitive and antagonistic in nature. Pseudomonas aeruginosa, a common Gram-negative soil bacterium and an opportunistic human pathogen, is well known for its ability to produce a blue phenazine, called pyocyanin, which is toxic to numerous bacteria and fungi. Recent reports indicate that P. Aeruginosa and Candida albicans can coexist in a variety of different opportunistic infections and a number of different molecular interactions between these two organisms have been described. Cabral et al.found that E. coli kills C. albicans when co-cultured in vitro. The study suggests that this activity results from a soluble factor produced by E. coli in a manner that is independent of the presence of fungal cells. They also found that magnesium limitation is required for the observed toxicity. Another interaction is that occurring between Staphyloococcus aureus and C.albicans, which although not yet fully characterized, appears to be synergistic. These complex interactions would have significant clinical implications if they occured in an immunocompromised host.Understanding the mechanisms of these interactions may lead to the development of novel antimicrobials.
3.It is important that the authors present the statistical results in table or tables, so that they are more understandable.
Answer : new figures are added (fig.2)
4. Explain why Fisher's test was used or chosen.5.The methodology section should be clearer in the development of all the processes that were carried.
Answer : Fisher's exact test is useful for contingency tables with very small samples size. Fisher's exact test determines whether a statistically significant association exists between two categorical variables.
6. Something interesting that the authors should take into account and explain in the results is why they used C. albicans and not other species of this genus. The latter is related to the fact that C. albicans is one of the two species that produce pseudohyphae and hyphae together with C. tropicalis. They should also explain in the methodology how they determined that the yeasts were C. albicans and not other species of the genus, since they mention Candida spp. in a section. Here they can preferably place the means used for identification (preferably Proteomics or Molecular Biology).
Answer: C.albicans is the most common pathogen in the majority clinical settings in our geographic area.
7. The authors should explain figure No. 5 in a clearer way.
8. In Figure 6, arrows should be used to indicate where the C. albicans yeasts are and where the P. aeruginosa bacteria are.
9. Both figures 7 and 9 are of very poor quality, it has a bad resolution or bad focus, so I suggest to improve or change it.
Answer: the figures have been improved and some of them had been changed.
10. It is not clear to me the objective of the study, at the beginning of the article.
Answer: The purpose of the study has been placed at the end of the introduction.
Our study investigated the impact of Gram-negative bacilli on the hyphal growth of C. albicans.
11. I understand that in the discussion they talk about the importance of the inhibition of C. albicans by compounds or metabolites released by Gram-negative bacilli (E. coli and P. aeruginosa), but this is not specifically mentioned in the introduction and conclusions, so I suggest improving them.
Answer: These aspects had been improved.The data inserted are mentioned in answer 1.
I appreciate your comments. Thank you for your time!
Round 2
Reviewer 1 Report
Comments and Suggestions for Authors
Improve the quality of the figures some of them are out of focus.
Author Response
In the current version, the research design, in particular introduction and discussion have been improved. New references are also added. I brought more examples from scientific literature on the interaction between fungi and bacteria. I have also analysed interaction between Pseudomonas aeruginosa and C. albicans. (possible pathophysiology of their interaction).
Regarding figures, unfortunately, is the only resolution I had.
Answer : In consequence, new data are added.
Introduction:
Brand et al. showed that P. aeruginosa cells kill C. albicans hyphal cells by creating biofilms, but not C. albicans yeast cells. They also investigated whether components of the hypha cell wall influenced susceptibility to the bacterium. It is well known that hypha-specific cell wall proteins (mannan components) of C. albicans are involved in adhesion and aggregation. His study demonstrated that mutant C. albicans strains that lacked the hypha-specific proteins Hyr1p, Hwp1p and Als3p or enzymes involved in N-glycosylation (Och1p, Mnn4p, Mnt3p, Mnt4p, and Mnt5p) of surface glycoproteins, were characterized by altered rates of killing by P. aeruginosa.(2)
In several studies of direct interactions between Pseudomonas spp. and a C. albicans monomorphic tup1mutant that is constitutively hyphal, P. aeruginosa was found to form a biofilm on hyphae and to selectively kill them.(3,4,5) Suppression of fungal growth has been also correlated with production of the bacterial phenazine derivatives, pyocynanin and 1-hydroxyphenazine, in culture media. Recent studies have identified N-(3-oxododecanoyl)-l-homoserine lactone (HSL) as a primary quorum-sensing molecule in P. aeruginosa. This molecule plays a crucial role in regulating bacterial virulence factor production and has been shown to inhibit hyphal development, reduce biofilm formation, and induce apoptosis in C.albicans.(6) These complex interactions would have significant clinical implications if they occured in an immunocompromised host. Understanding the mechanisms of these interactions may lead to the development of novel antimicrobials.
Discussion:
Brand et al. showed that P. aeruginosa cells kill C. albicans hyphal cells by creating biofilms, but not C. albicans yeast cells .They tested a range of mutants that lacked hypha-specific cell wall mannoproteins and others that lacked specific glycosyl epitopes. His study demonstrated that mutant C. albicans strains that lacked the hypha-specific proteins Hyr1p, Hwp1p and Als3p or enzymes involved in N-glycosylation (Och1p, Mnn4p, Mnt3p, Mnt4p, and Mnt5p) of surface glycoproteins, were characterized by altered rates of killing by P. aeruginosa.
However, the survival of mnt1Δ, mnt2Δ and mnt1Δ/mnt2Δ mutants with truncated O-linked mannan was significantly reduced in the presence of P. aeruginosa as compared with the control strain, suggesting that O-mannan is protective against the P. aeruginosa killing activity. P. aeruginosa adhered preferentially to specific hyphae at all time points. There appeared to be no preferred attachment site relative to the length of the hypha, the presence or absence of a branch or any other morphological parameter.(2)
Other factor influencing in vitro interaction is iron. A large proportion of the increased protein production such as pyoverdine was attributed to a siderophore, pyoverdine, specific to P. aeruginosa. This increase in pyoverdine is thought to be due to the increased iron utilization in the mixed biofilm. This was confirmed by the addition of iron, which abolished the production of pyoverdine. It was demonstrated that sequestration of available iron by pyoverdine results in decreased availability to C. albicans, although C. albicans is able to utilize iron bound to certain other microbial siderophores. Other evidence suggests that this phenomenon may not be of importance during in vivo interaction (in murine models). The authors suspect the heterogeneity of the biofilms between in vivo and in vitro studies may cause the differential results. They also found that hypoxia influences the ability of P. aeruginosa to inhibit C. albicans filamentation in vitro compared to aerobic conditions. This was attributed to decreased AHL produced by P. aeruginosa in the presence of C. albicans. Additionally, the authors also speculated that competition for iron may also be greater during hypoxia Therefore, both the interaction of P. aeruginosa with C. albicans, the concentration of oxygen and iron competition influences the production of HSL.(6)
I appreciate your comments. Thank you for your time!
Reviewer 2 Report
Comments and Suggestions for Authors
1: The Author has revised the manuscript; however, some of the comments are still not addressed satisfactorily
such as comment 4. The methodology is not detailed or reproducible. Since Candida albicans typically exhibits an antagonistic relationship with Pseudomonas aeruginosa, this interaction should be discussed by the author.
2: Many sentences in the introduction and discussion sections need to add references.
3: Check line 181>>>>>: Figure 3-11 ??
Author Response
Reviewer 2:
In the current version, the research design, in particular introduction and discussion have been improved. New references are also added. I brought more examples from scientific literature on the interaction between fungi and bacteria. I have also analysed interaction between Pseudomonas aeruginosa and C. albicans. (possible pathophysiology of their interaction).
Line 181 has been resolved.
Answer : In consequence, new data are added.
Introduction:
Brand et al. showed that P. aeruginosa cells kill C. albicans hyphal cells by creating biofilms, but not C. albicans yeast cells. They also investigated whether components of the hypha cell wall influenced susceptibility to the bacterium. It is well known that hypha-specific cell wall proteins (mannan components) of C. albicans are involved in adhesion and aggregation. His study demonstrated that mutant C. albicans strains that lacked the hypha-specific proteins Hyr1p, Hwp1p and Als3p or enzymes involved in N-glycosylation (Och1p, Mnn4p, Mnt3p, Mnt4p, and Mnt5p) of surface glycoproteins, were characterized by altered rates of killing by P. aeruginosa.(2)
In several studies of direct interactions between Pseudomonas spp. and a C. albicans monomorphic tup1mutant that is constitutively hyphal, P. aeruginosa was found to form a biofilm on hyphae and to selectively kill them.(3,4,5) Suppression of fungal growth has been also correlated with production of the bacterial phenazine derivatives, pyocynanin and 1-hydroxyphenazine, in culture media. Recent studies have identified N-(3-oxododecanoyl)-l-homoserine lactone (HSL) as a primary quorum-sensing molecule in P. aeruginosa. This molecule plays a crucial role in regulating bacterial virulence factor production and has been shown to inhibit hyphal development, reduce biofilm formation, and induce apoptosis in C.albicans.(6) These complex interactions would have significant clinical implications if they occured in an immunocompromised host. Understanding the mechanisms of these interactions may lead to the development of novel antimicrobials.
Discussion:
Brand et al. showed that P. aeruginosa cells kill C. albicans hyphal cells by creating biofilms, but not C. albicans yeast cells .They tested a range of mutants that lacked hypha-specific cell wall mannoproteins and others that lacked specific glycosyl epitopes. His study demonstrated that mutant C. albicans strains that lacked the hypha-specific proteins Hyr1p, Hwp1p and Als3p or enzymes involved in N-glycosylation (Och1p, Mnn4p, Mnt3p, Mnt4p, and Mnt5p) of surface glycoproteins, were characterized by altered rates of killing by P. aeruginosa.
However, the survival of mnt1Δ, mnt2Δ and mnt1Δ/mnt2Δ mutants with truncated O-linked mannan was significantly reduced in the presence of P. aeruginosa as compared with the control strain, suggesting that O-mannan is protective against the P. aeruginosa killing activity. P. aeruginosa adhered preferentially to specific hyphae at all time points. There appeared to be no preferred attachment site relative to the length of the hypha, the presence or absence of a branch or any other morphological parameter.(2)
Other factor influencing in vitro interaction is iron. A large proportion of the increased protein production such as pyoverdine was attributed to a siderophore, pyoverdine, specific to P. aeruginosa. This increase in pyoverdine is thought to be due to the increased iron utilization in the mixed biofilm. This was confirmed by the addition of iron, which abolished the production of pyoverdine. It was demonstrated that sequestration of available iron by pyoverdine results in decreased availability to C. albicans, although C. albicans is able to utilize iron bound to certain other microbial siderophores. Other evidence suggests that this phenomenon may not be of importance during in vivo interaction (in murine models). The authors suspect the heterogeneity of the biofilms between in vivo and in vitro studies may cause the differential results. They also found that hypoxia influences the ability of P. aeruginosa to inhibit C. albicans filamentation in vitro compared to aerobic conditions. This was attributed to decreased AHL produced by P. aeruginosa in the presence of C. albicans. Additionally, the authors also speculated that competition for iron may also be greater during hypoxia Therefore, both the interaction of P. aeruginosa with C. albicans, the concentration of oxygen and iron competition influences the production of HSL.(6)
I appreciate your comments. Thank you for your time.
Reviewer 3 Report
Comments and Suggestions for Authors
despite my suggestions the authors only added general concern and some biological results obtained previously and did not explain what are the clinical consequences and possible pathophysiology of candida/bacterial interactions.
Author Response
Reviewer 3:
In the current version, the research design, in particular introduction and discussion have been improved. New references are also added. I brought more examples from scientific literature on the interaction between fungi and bacteria. I have also analysed interaction between Pseudomonas aeruginosa and C. albicans. (possible pathophysiology of their interaction).
Answer : In consequence, new data are added.
Introduction:
Brand et al. showed that P. aeruginosa cells kill C. albicans hyphal cells by creating biofilms, but not C. albicans yeast cells. They also investigated whether components of the hypha cell wall influenced susceptibility to the bacterium. It is well known that hypha-specific cell wall proteins (mannan components) of C. albicans are involved in adhesion and aggregation. His study demonstrated that mutant C. albicans strains that lacked the hypha-specific proteins Hyr1p, Hwp1p and Als3p or enzymes involved in N-glycosylation (Och1p, Mnn4p, Mnt3p, Mnt4p, and Mnt5p) of surface glycoproteins, were characterized by altered rates of killing by P. aeruginosa.(2)
In several studies of direct interactions between Pseudomonas spp. and a C. albicans monomorphic tup1mutant that is constitutively hyphal, P. aeruginosa was found to form a biofilm on hyphae and to selectively kill them.(3,4,5) Suppression of fungal growth has been also correlated with production of the bacterial phenazine derivatives, pyocynanin and 1-hydroxyphenazine, in culture media. Recent studies have identified N-(3-oxododecanoyl)-l-homoserine lactone (HSL) as a primary quorum-sensing molecule in P. aeruginosa. This molecule plays a crucial role in regulating bacterial virulence factor production and has been shown to inhibit hyphal development, reduce biofilm formation, and induce apoptosis in C.albicans.(6) These complex interactions would have significant clinical implications if they occured in an immunocompromised host. Understanding the mechanisms of these interactions may lead to the development of novel antimicrobials.
Discussion:
Brand et al. showed that P. aeruginosa cells kill C. albicans hyphal cells by creating biofilms, but not C. albicans yeast cells .They tested a range of mutants that lacked hypha-specific cell wall mannoproteins and others that lacked specific glycosyl epitopes. His study demonstrated that mutant C. albicans strains that lacked the hypha-specific proteins Hyr1p, Hwp1p and Als3p or enzymes involved in N-glycosylation (Och1p, Mnn4p, Mnt3p, Mnt4p, and Mnt5p) of surface glycoproteins, were characterized by altered rates of killing by P. aeruginosa.
However, the survival of mnt1Δ, mnt2Δ and mnt1Δ/mnt2Δ mutants with truncated O-linked mannan was significantly reduced in the presence of P. aeruginosa as compared with the control strain, suggesting that O-mannan is protective against the P. aeruginosa killing activity. P. aeruginosa adhered preferentially to specific hyphae at all time points. There appeared to be no preferred attachment site relative to the length of the hypha, the presence or absence of a branch or any other morphological parameter.(2)
Other factor influencing in vitro interaction is iron. A large proportion of the increased protein production such as pyoverdine was attributed to a siderophore, pyoverdine, specific to P. aeruginosa. This increase in pyoverdine is thought to be due to the increased iron utilization in the mixed biofilm. This was confirmed by the addition of iron, which abolished the production of pyoverdine. It was demonstrated that sequestration of available iron by pyoverdine results in decreased availability to C. albicans, although C. albicans is able to utilize iron bound to certain other microbial siderophores. Other evidence suggests that this phenomenon may not be of importance during in vivo interaction (in murine models). The authors suspect the heterogeneity of the biofilms between in vivo and in vitro studies may cause the differential results. They also found that hypoxia influences the ability of P. aeruginosa to inhibit C. albicans filamentation in vitro compared to aerobic conditions. This was attributed to decreased AHL produced by P. aeruginosa in the presence of C. albicans. Additionally, the authors also speculated that competition for iron may also be greater during hypoxia Therefore, both the interaction of P. aeruginosa with C. albicans, the concentration of oxygen and iron competition influences the production of HSL.(6)
*
I appreciate your comments. Thank you for your time!
All the best, Dr. Madalina Bordea.
Reviewer 4 Report
Comments and Suggestions for Authors
The authors complied with the corrections made, so I consider that the article can be published in the current form.
Author Response
I appreciate your comments. Thank you for your time!
Round 3
Reviewer 1 Report
Comments and Suggestions for Authors
Figs. 5,6 &* are in low quality please improve.
There is a gap between Fig. 5 and 6 with the following Legends without the figure:
"B) Test group images, Candida albicans and either E. coli or P. aeruginosa polymicrobial culture:
Reviewer 3 Report
Comments and Suggestions for Authors
no further comments